# Prevention of Secondary Caries Using Resin-Based Pit and Fissure Sealants Containing Hydrated Calcium Silicate

**DOI:** 10.3390/polym12051200

**Published:** 2020-05-25

**Authors:** Song-Yi Yang, Ji-Won Choi, Kwang-Mahn Kim, Jae-Sung Kwon

**Affiliations:** 1Department and Research Institute of Dental Biomaterials and Bioengineering, Yonsei University College of Dentistry, Seoul 03722, Korea; syyang88@yuhs.ac (S.-Y.Y.); den0424@yuhs.ac (J.-W.C.); kmkim@yuhs.ac (K.-M.K.); 2BK21 PLUS Project, Yonsei University College of Dentistry, Seoul 03722, Korea

**Keywords:** hydrated calcium silicate, pit and fissure sealant, caries inhibition, acid-neutralizing property, calcium ion release

## Abstract

The objective of this study was to investigate the effects of hydrated calcium silicate filler (hCS) on resin-based pit and fissure sealants’ acid neutralization, calcium ion release, and mechanical and physical properties. To produce the hCS filler, Portland cement (CS) was mixed with distilled water and ground into fine particles. The particles were then mixed with silanized glass filler and added to a photo-activated resin matrix. To evaluate the acid neutralization and calcium ion release properties, the specimens were immersed in a pH 4.0 lactic acid solution and distilled water for 28 days. Also, the flexural strength, depth of cure, water sorption, and solubility were tested. All of the groups containing hCS and CS required less than one minute to increase the pH from 4.0 to 5.5. With 50% hCS, the calcium ion release was higher than 50% CS in the distilled water at the initial time. The flexural strength and depth of cure decreased according to the increasing proportion of hCS added. The water sorption and solubility had an increasing trend as increasing proportions of hCS were added. These findings showed that pit and fissure sealant containing hCS exhibit superior acid neutralization and calcium release properties, and may be promising for caries-inhibiting dental material.

## 1. Introduction

Dental caries is a chronic oral disease that influences most populations and is considered harmful to global oral health [1]. Although children and adolescents have healthier teeth than in the past, approximately one-fourth of children and more than half of adolescents experience dental caries in their permanent teeth, which remains a major oral disease [2]. Pits and fissures comprise only 12.5% of tooth surfaces but account for 88% of caries in children [3].

Several studies assessed various treatments for high-risk patients exposed to dental caries including mechanical plaque control using brushing, chemical treatment of acid-resistant tooth structures via the application of fluoridate supplements, and physical sealing of pits and fissures using sealants [4,5,6]. Pit and fissure sealants for children and adolescents have been demonstrated and applied in a relatively safe and effective manner for more than 60 years [7].

However, pit and fissure sealants may not reach the bottom of pits and fissures and hence a space remains under the polymerized sealant material [8]. Secondary caries or recurrent caries often occurs around sealed pits and fissures between the tooth structure and material due to either microleakage induced by the materials’ polymerization shrinkage or partial detachment from the tooth [9,10,11].

To overcome these problems, bioactive dental composite materials have been developed to prevent dental caries [8,12,13,14]. In a previous study, pit and fissure sealants containing alkaline bioactive glass filler neutralized an acidic environment and were effective for preventing tooth demineralization of materials around the area [15,16]. In another study, amorphous calcium phosphate filler incorporated into composite resin materials released supersaturating calcium and phosphate ions in an aqueous environment, effectively remineralizing enamel lesions [17,18,19,20,21]. Many studies have focused on only acid neutralization or calcium and phosphate releasing properties to inhibit dental caries, which prevented demineralization and simultaneously promoted remineralization. Therefore, a bioactive dental composite material with continuous acid neutralization and calcium ion release that can be effectively utilized in oral environments in patients at a high risk of exposure to dental caries must be developed.

Portland cement as a mineral trioxide aggregate (MTA) is composed of calcium silicate complex. Portland cement is currently a popular choice of dental material because of its biocompatibility and bioactive properties [22]. It has been reported that this calcium silicate promotes the formation of an alkaline pH and releases high levels of calcium ions in aqueous environments [23]. Hydrated calcium silicate crystals penetrate dentinal tubules and are acid resistant, which are good characteristics for dentine hypersensitivity therapy [24,25,26,27]. This study used hydrated calcium silicate derived from Portland cement as a bioactive pit and fissure sealant filler for patients at a high risk of exposure to dental caries. Hydrated calcium silicate is added to pit and fissure sealants because it reacts with oral fluids for acid neutralization and remineralization.

To date, there has been no evidence in the literature of the caries-inhibition properties of pit and fissure sealants with hydrated calcium silicate materials for acid neutralization and remineralization. Therefore, pit and fissure sealants with hydrated calcium silicate are promising for the prevention of secondary caries with carcinogenicity.

Accordingly, the purpose of this study was to (1) evaluate the neutralizing effects and (2) examine the calcium ion release of caries-inhibiting capabilities under cariogenic challenge with possible applications as hydrated calcium silicate material composites based on Portland cement as dental pit and fissure sealants. The null hypothesis of this study was that (1) pit and fissure sealants containing hydrated calcium silicate filler would not result in significant differences in the acid-neutralizing properties compared to those of pit and fissure sealants without hydrated calcium silicate filler; (2) pit and fissure sealants containing hydrated calcium silicate filler would not result in significant differences in calcium ion release compared to those of pit and fissure sealants without hydrated calcium silicate filler; and (3) pit and fissure sealants containing hydrated calcium silicate filler would not result in significant differences in mechanical and physical properties compared to those of pit and fissure sealants without hydrated calcium silicate filler.

## 2. Materials and Methods

### 2.1. Preparation of Hydrated Calcium Silicate Cement Powder

For the experiments, a commercially available white Portland cement (Union White Portland cement, Gyunggido, Korea) was used as the calcium silicate cement (CS). To produce the hydrated calcium silicate cement (hCS), CS was dispensed on a mixing pad and gradually mixed with distilled water in a 0.33 water/powder ratio for 1 min as previously recommended [25]. The mixed paste consisting of CS powder and distilled water was stored at 100% relative humidity and (37 ± 1) °C for 3 days. After the hCS set, stable solid powder was obtained using a mortar and pestle, followed by grinding with a planetary ball mill (PM 100 CM, Retsch, Haan, Germany) at 450 rpm for 40 min. The ground hCS powder was filtered through a 400-mesh sieve shaker (AS200, Retsch, Haan, Germany) to obtain fine particles less than 38 µm in size. They were stored in a desiccator at room temperature until further use.

### 2.2. Characterization of the Experimental hCS Powder

The characterization of the experimental powders was verified both before (CS) and after hydration in distilled water (hCS) according to the following techniques. The experimental powder phase analysis was performed using X-ray diffraction (XRD, Ultima IV, Rigaku, Tokyo, Japan). A 2θ angle range of 10° to 60° was used with a scanning speed of 2°/min. To confirm the appearance of the experimental powders, small amounts of CS and hCS powder were attached to metal stubs via double adhesive carbon tape, gold-palladium (Au-Pd) sputter coated in a vacuum for 90 s at 20 mA, and examined by scanning electron microscopy (SEM) (JEOL-7800F, JEOL Ltd., Tokyo, Japan) at a magnification of 2000×. The particle size distribution of CS and hCS powder was determined using a particle size analyzer (Mastersizer 2000, Malvern, Malvern, Worcestershire, UK) with distilled water as a dispersion medium at a rotating speed at 2000 rpm to prevent the agglomerated powder from sinking to the bottom. The time from placing the experimental powder into the dispersion medium until analysis was less than 10 s.

### 2.3. Preparation of Pit and Fissure Sealants Containing hCS

A light-polymerizable resin matrix was prepared by blending 49.5 wt % bisphenol A glycerolate dimethacrylate (Bis-GMA, Sigma-Aldrich, Steinheim, Germany), 49.5 wt % triethylene glycol dimethacrylate (TEGDMA, Sigma-Aldrich, Steinheim, Germany), 0.3 wt % camphorquinone (CQ, Sigma-Aldrich, Steinheim, Germany), and 0.6 wt % 2-(dimethylamino)ethyl methacrylate (Sigma-Aldrich, Steinheim, Germany). Conventional silanized dental glass filler (180 ± 30 nm; NanoFine NF180, Schott, Landshut, Germany) was obtained from a commercial source. Experimental powders and conventional silanized filler were added to the resin matrix and mixed using a magnetic stirrer in the dark for one day to obtain a homogeneous state. Using this method, six experimental pit and fissure sealant groups were prepared with varying experimental powders and silanized dental glass filler proportions (Table 1).

### 2.4. Acid-Neutralizing Properties

Uncured experimental pit and fissure sealant materials were dispensed in a stainless-steel mold with dimensions of 25 mm × 2 mm × 2 mm and clamped between slide glasses. The uncured experimental pit and fissure sealant material was immediately polymerized with overlapping areas for 20 s using an LED light-curing unit (Elipar S10, 3M ESPE Co., Seefeld, Germany). After the experimental pit and fissure sealant material polymerized, a bar-shaped cured specimen was separated from the stainless-steel mold and immersed in lactic acid solution (pH 4.0) simulating a cariogenic condition, yielding a specimen volume per solution ratio of 0.14 cm^3^/1 mL, the same as in a previous study [15,21]. Before immersion of the specimens, the pH of the acid solution was adjusted to 4.0 by dropping a small amount of lactic acid (Sigma-Aldrich, Steinheim, Germany) into a beaker full of distilled water at (25 ± 1) °C.

To evaluate the initial acid-neutralizing properties, immediately after immersion, the lactic acid solution’s pH change with the experimental specimens were acquired each minute using a digital pH-meter (Orion 4 Star, Thermo Fisher Scientific, Singapore) that had been calibrated at pH 4.01, pH 7.0, and pH 10.01. The time for the solution’s pH to increase from 4.0 to 5.5 was recorded for 120 min.

To confirm the prolonged acid-neutralizing properties, measurements were obtained at periods up to 28 days. Following the evaluation of the initial acid-neutralizing properties, the specimens were dried with a laboratory tissue and transferred into a pH 4.0 fresh lactic acid solution, yielding a specimen volume/solution ratio of 0.14 cm^3^ to 1 mL. This was repeated at 1, 2, 4, 7, 14, 21, and 28 days to simulate the accumulated acidic cariogenic challenge. Each time, after the specimens were removed, the collected storage solutions were recorded using a digital pH meter.

### 2.5. Calcium Ion Release

To analyze the calcium ion concentration released from the specimens, bar-shaped samples (25 mm × 2 mm × 2 mm) were obtained using previously described procedures and immersed in lactic acid solution with pH 4.0 and distilled water (JW pharmaceutical, Seoul, Korea) with pH 6.5, respectively, at (37 ± 1) °C for the duration of the experiment. Herein, 0.14 cm^3^ of sample volume was available for calcium ion release per one milliliter of two types of immersion solution. The specimens were immersed for 2 h, consecutively transferred into a fresh immersion solution after 1, 2, 4, 7, 14, 21, and 28 days. Each time, after the specimens were removed, the collected storage solutions were analyzed for their calcium ion concentrations using a calcium ion selective electrode (Thermo Fisher Scientific, Waltham, MA, USA) that was previously calibrated with 1 mM, 10 mM, and 100 mM of Ca/L standard solutions.

### 2.6. Three-Point Flexural Strength

To confirm the mechanical properties, the flexural strength was measured using the three-point flexural strength method outlined in ISO 4049 (2019) for polymer-based dental restorative materials. The bar-shaped specimens (25 mm × 2 mm × 2 mm) were prepared using a previously described method and immersed in distilled water at (37 ± 1) °C for 24 h. The cured specimen was fractured using a computer-controlled universal testing machine (Instron 5942, Instron, Norwood, MA, USA) at a crosshead speed of 1 mm/min. The maximum load (*F*) was recorded and the three-point flexural strength (S) was calculated using following equation: S = 3*Fl*/(2*bh*^2^), where *F* is the maximum fracture load, *l* is the 20 mm support distance, *b* is the specimen width, and *h* is the specimen height.

### 2.7. Depth of Cure

The depth of cure of the experimental pit and fissure sealant was determined following ISO 6874 (2015), a standardized technique for polymer-based pit and fissure dental sealant materials. A cylindrical metallic mold (6 mm long and 4 mm in diameter) was placed on a microscope slide glass covered with a polyester film. The mold was filled with uncured test materials, avoiding the formation of air bubbles. The top of the mold was then covered with a polyester film and irradiated for 20 s using LED light curing from the top. Immediately after irradiation, the specimen was separated from the mold and the uncured soft material was scraped away with a plastic spatula. The height of the cured solid material was measured at three different points using a digital micrometer (Mitutoyo, Japan) with an accuracy of 0.1 mm. The mean was calculated as the depth of cure of each specimen.

### 2.8. Water Sorption and Solubility

The water sorption and solubility of the pit and fissure sealant was conducted according to ISO 4049 (2019). Cured disk specimens (diameter 15.0 ± 0.1 mm, thickness 1.0 ± 0.1 mm) were stored in a desiccator maintained at (37 ± 1) °C. After 22 h, the specimens were transferred and stored in a desiccator maintained at (23 ± 1) °C for 2 h and then weighed in a digital balance (XS105, Mettler-Toledo AG, Greifensee, Switzerland) with an accuracy of 0.1 mg until a constant mass (*m*_1_) was obtained. The diameter and thickness of the specimens were measured using a digital micrometer. The values were used to calculate the volume (*V*) of each sample (in 0.01 mm^3^). The specimens were then individually immersed in distilled water at (37 ± 1) °C for 7 days. After 7 days, each specimen’s visible moisture was blotted dry with tissue paper and weighed for mass (*m*_2_) again. Each specimen was stored in a desiccator and reweighed daily until a constant dry mass (*m*_3_) was obtained. The water sorption and solubility were calculated using the following equations: W_sp_ = (*m*_2_
*− m*_3_)/*V* and W_sl_ = (*m*_1_
*− m*_3_)/*V*, where W_sp_ is the test material’s absorption (μg/mm^3^) and W_sl_ is its solubility (μg/mm^3^).

### 2.9. Statistical Analysis

All of the data on the acid-neutralizing properties, Ca ion release, and mechanical and physical properties were analyzed with one-way ANOVA (SPSS 25, IBM Co., Armonk, NY, USA) to confirm the interactions between factors (experimental powder content and immersion time). To assess these factors’ significant differences, Tukey’s statistical test was applied at a significance level of 0.05.

## 3. Results

### 3.1. Characterization of the Experimental Powders

The XRD patterns of the CS and hCS powders are presented in Figure 1. Both had major tricalcium silicate peaks (C3S) at 2θ = 29.34°, 32.14°, and 51.64° and peaks corresponding to bicalcium silicate (C2S, 2θ = 32.48° and 41.22°) and tetracalcium aluminoferrite (C4AF, 2θ = 34.26°). However, the peaks’ relative line intensity was lower in the hCS powder than the CS powder. The hCS powder had a sharp calcium hydroxide (CH) peak at 2θ = 18°, which was not observed in the CS powder. Figure 2 shows the micromorphology of the CS and hCS powders. Before hydration (Figure 2a), the CS powders markedly varied in particle size below 25 μm and the particles had irregular and sharp-edged shapes. After hydration (Figure 2b), the hCS powders also had a wide distribution of particle sizes but appeared blunt-shaped in morphology and highly agglomerated as the smaller particles tended to adhere to the surfaces of the larger particles. The results of the particle size analysis and the micron size of the powders are presented in Figure 3. In terms of the CS powder’s particle size distribution, D10 (10% of the particles were smaller), D50 (median particle size), and D90 (10% of particles were smaller) were 4.36 μm, 16.12 μm, and 39.72 μm, respectively. For the hCS powder, D10, D50, and D90 were 3.57 μm, 18.53 μm, and 38.63 μm, respectively.

### 3.2. Acid-Neutralizing Properties

The results of the initial pH change are shown in Figure 4a. The addition of hCS filler rapidly increased the pH in the first minutes immediately after immersion, followed by incremental increases in pH during 120 min of storage for the hCS filler containing groups compared to the hCS 0 group, which had no hCS filler. The pH value of the solution with CS 50.0 had an increased pH for 120 min after immersion, as both had a pH over 10.0 after 120 min. These final pH values were both significantly higher than the hCS 0 value (*p* < 0.05). All of the groups containing hCS and CS required less than one minute for the pH to increase from 4.0 to 5.5. However, the hCS 0 group’s pH did not increase from 4.0 to 5.5 until 120 min. The acid-neutralizing ability of the experimental groups over periods of time are plotted in Figure 4b. The pH of the solution with CS and hCS containing groups was significantly higher than the pH of the hCS 0 group throughout the test time (*p* < 0.05). Each group containing from 12.5% to 50.0% hCS and CS filler had increased final pH values as more hCS and CS filler was added. After 28 days, the pH of the solution with hCS 37.5, hCS 50.0, and CS 50.0, was 11.30 ± 0.05, 11.99 ± 0.19, and 11.67 ± 0.05, respectively. The final pH of hCS 50.0 was not significantly different from the value of CS 50.0 (*p* > 0.05). The results of hCS 12.5 and hCS 25.0 had decreased trends compared to hCS 37.5, hCS 50.0, and CS 50.0 after 1 day, but they were still alkaline until the end of the experiment. The pH was 8.03 ± 0.19 and 10.02 ± 0.14, respectively (*p* < 0.05). In comparison, the pH of the hCS 0 group was significantly lower than the CS and hCS containing groups at all of the experimental times (*p* < 0.05). The pH increased slightly for 2 h, but then remained at approximately 4 despite prolonged immersion.

### 3.3. Calcium Ion Release

The ion release results in terms of the concentration of leached calcium ion for 2 h, 1, 2, 4, 7, 14, 21, and 28 days in distilled water and lactic acid solution are shown in Figure 5. In the distilled water, hCS 37.5, hCS 50.0, and CS 50.0 showed burst calcium ion release until day 1 and decreased significantly over the immersion time as shown in Figure 5a. The initial calcium ion concentration of hCS 50.0 was significantly higher than the other groups (*p* < 0.05) and 5-fold that of the non-hydrated group for CS 50.0. However, the amount of released calcium from hCS 12.5 was nearly zero until day 14 and a small amount of ions was released from 14 to 28 days. At the end of 28 days of immersion, there were no significant differences between hCS 12.5 and hCS 0. For the calcium ion released in lactic acid solution (Figure 5b), hCS 37.5 and hCS 50.0 showed burst calcium ion release until day 1, and then the trend decreased over the immersion time. In contrast, CS 50.0 initially demonstrated a low amount of calcium ion release until day 2, and then a gradual increase until day 14. The hCS 37.5 and hCS 50.0 groups had lower initial calcium ion release in the lactic acid solution than in the distilled water (*p* < 0.05), indicating that the distilled water promoted the initial calcium ion release.

### 3.4. Three-Point Flexural Strength

The three-point flexural strength test results are shown in Figure 6a. Increasing the hCS filler level while decreasing the silanized glass filler level significantly decreased the flexural strength (*p* < 0.05). The flexural strength was significantly affected by the incorporation of 37.5% or more of hCS. In contrast, the samples containing less than 25.0% had similar flexural strength to the hCS 0 control group. The hCS 25.0 and hCS 12.5 groups were not significantly different from 58.64 ± 3.43 MPa in hCS 0 (*p* > 0.05). CS 50.0, which had only non-hydrated CS powder as a filler, had flexural strength of 20.03 ± 2.03 MPa, significantly higher than 5.84 ± 0.89 MPa in the hCS 50.0 group (*p* < 0.05).

### 3.5. Depth of Cure

The depth of cure results are shown in Figure 6b. The depth of cure decreased as an increasing proportion of hCS filler was added. The results of the depth of cure were significantly different among the groups (*p* < 0.05), except for hCS 0 and hCS 12.5, which did not significantly differ (*p* > 0.05). hCS 50.0 and CS 50.0 had cure depths of 3.07 ± 0.08 mm and 2.65 ± 0.38 mm, respectively, significantly lower than 5.99 ± 0.01 mm in the hCS 0 control group (*p* < 0.05). Nevertheless, all of the groups’ values were more than 1.5 mm and fulfilled the requirement of ISO 6874.

### 3.6. Water Sorption and Solubility

The water sorption and solubility results are shown in Figure 6c,d. Among the different groups, there were significant differences in both the water sorption and solubility (*p* < 0.05). The water sorption and solubility increased as an increasing proportion of hCS was added (*p* < 0.05). hCS 50.0 had the highest value of both of these properties. The water sorption and solubility of CS 50.0 was significantly lower than that of hCS 50.0 (*p* < 0.05).

## 4. Discussion

To the best of our knowledge, this is the first study on the development of a pit and fissure sealant containing hydrated calcium silicate filler to prevent dental caries. The combination of resin matrix with hydrated calcium silicate filler resulted in superior acid neutralization, high calcium ion release, and sustained effects for a prolonged time period.

The materials’ XRD patterns provided chemical, phase, and crystal structure data necessary to understand experimental powder performance [28]. The XRD results showed that the major constituents in the CS and hCS powders were C3S, C2S, and C4AF. Thus, CS and hCS have similar major constituents. CH was observed only in hCS and not in CS in accordance with previous reports on these materials [29,30]. Portland cement-based materials have been studied as desensitizing agents due to their bioavailable calcium component [26,27] and are proposed for caries-preventing material such as functional bioactive filler. This is the first study reporting the anti-caries efficacy of a test material using Portland cement as a functional filler in resin composite.

Many studies in the literature have reported the presence of acidogenic oral bacteria in marginal gaps between the tooth structure and composite material, and these gaps are highly susceptible to dental caries [31,32,33]. Oral bacteria ferments carbohydrates and produces various organic acids including lactic acid, formic acid, propionic acid, and acetic acids, which dissolve the calcium and phosphate ions from the tooth structure [31]. Although saliva has buffering capacity and contains trace calcium and phosphate ions, the acid neutralization of local plaque pH and remineralization of demineralized tooth lesions can be significantly improved by increasing the level of calcium ions in the surrounding solution much higher than in natural oral fluid [33]. Therefore, developing bioactive pit and fissure sealant material is important for demineralization and promoting remineralization. Also, secondary or recurrent caries do not develop immediately after pit and fissure sealant is applied to the occlusal surface; the material must maintain bioactive properties over prolonged periods of time [34,35]. However, there are few studies on the prolonged acid neutralization capabilities of bioactive dental composite materials, one way to prevent the progression of caries.

The first null hypothesis—pit and fissure sealants containing hydrated calcium silicate filler would not result in significant differences in the acid-neutralizing properties compared to those of pit and fissure sealants without hydrated calcium silicate filler—was rejected. Our study demonstrated that hydrated and non-hydrated calcium silicate composite materials rapidly increased the pH of lactic acid solution. As soon as the experimental specimens, except for hCS 0, were immersed in lactic acid solution, the pH immediately increased to above 5.5 in the first minute, and then stabilized at pH 10~11 for a short period of time until 120 min. The critical pH level in the mouth is 5.5; when the pH drops below this point, the tooth structure starts to demineralize or dissolve [15,21,36,37]. It can be adapted for a material to quickly increase the cariogenic pH from 4.0 to above 5.5 to prevent tooth demineralization. Rapidly increasing the pH of local dental plaque from a cariogenic level to a safe level plays an important role in preventing tooth structures from dissolving.

The second null hypothesis—pit and fissure sealants containing hydrated calcium silicate filler would not result in significant differences in calcium ion release compared to those of pit and fissure sealants without hydrated calcium silicate filler—was rejected. In the present study, the experimental groups with a higher amount of hydrated calcium silicate filler had higher initial calcium ion release. These pit and fissure sealants containing hydrated calcium silicate filler occupy sites with a high risk of exposure to dental caries compared to pit and fissure sealants containing non-hydrated calcium silicate filler. Calcium silicate-based materials have been developed as root-end filling materials and proposed for additional clinical applications such as pulp capping and perforation repair [27,38]. Several studies reported that the hydration of calcium silicate-based material produces calcium hydroxide, causing the release of calcium ions and promoting alkaline pH in the presence of biological fluids [23,29,30].

Long-term high humidity, mechanical wear, occlusal stress, and complicated oral environments may cause the dissolution and partial loss of bioactive filler from the resin matrix of pit and fissure sealants [39]. Therefore, it is necessary to develop a pit and fissure sealant material for caries inhibition that is effective for acid neutralization and remineralization of demineralized enamel in an oral environment. The present study verified the ability of hCS and CS filler to maintain acid-neutralizing and calcium ion releasing properties throughout prolonged exposure to a lactic acid solution, and these properties increased as more hydrated calcium silicate filler was added. In prior studies, nano-sized amorphous calcium phosphate or micro-sized 45S5 bioactive glass were used as functional fillers to prevent secondary caries [15,18]. However, these studies required a longer time to neutralize the acid solution than our study and did not demonstrate continual caries prevention. In this study, the acid neutralization and calcium ion release test showed that the release produced a high concentration of calcium hydroxide and thus alkalinized the surrounding fluid during the hydration reaction. Based on these findings, hydrated calcium silicate material incorporated into resin matrix that can release calcium hydroxide ions should have more potential for a prolonged time and self-repair against demineralization at the material and tooth interface.

The third null hypothesis—pit and fissure sealants containing hydrated calcium silicate filler would not result in significant differences in mechanical and physical properties compared to those of pit and fissure sealants without hydrated calcium silicate filler—was partially accepted. The incorporation of hydrated or non-hydrated calcium silicate enables pit and fissure sealants to release calcium ions to prevent dental caries while silanized glass filler provides the required load-bearing ability [20]. In enlarged pit and fissure sealing, the material’s mechanical properties are important since the material can be placed in areas that undergo mechanical stresses during clenching [7]. The hCS 12.5 group had flexural strength matching hCS 0, which was a non-releasing pit and fissure sealant. In contrast, the strength of hCS 50.0 and CS 50.0 without silanized glass filler was significantly lower than hCS 0, similar to previous studies showing that ion releasing composite had flexural strengths half that of unfilled composite [19,40]. Consequently, long-term continuous ion release of hCS 50.0 and CS 50.0 is likely to result in deterioration of the mechanical properties compared with hCS 0. Although the factor of size difference between hCS and commercial filler cannot be ruled out, the silane treatment is an important factor causing the difference in resin composite strength. CS and hCS fillers did not reinforce the mechanical properties, likely because these materials were not silanized and hence could not bond chemically with the resin matrix. Silanization of calcium silicate fillers was not desirable because it reduced the acid neutralization capacity and calcium ion release. Therefore, a more effective approach to increasing the strength of pit and fissure sealants is using a strong filler that chemically and mechanically bonds with the resin matrix for reinforcement. In another way to improve the strength, smaller nano-sized particles are needed to fill the gap between the micro-sized hCS and the conventional silanized dental glass filler. Smaller nano-sized particles would act as a filler to prevent the occurrence of pits, and nanoparticle-reinforced rein composites help to enhance the mechanical properties of material [41,42]. In a previous study, the flexural strength of a commercially available pit and fissure sealant was above approximately 40 MPa [7]. This matched or exceeded the strength of hCS 12.5 and 25.0 in the present study, suggesting that these might be useful pit and fissure sites where sealant material is placed.

Good polymerization properties of pit and fissure sealants are closely related to successful clinical sealing of occlusal pits and fissures, especially when they are narrow and deep [43]. As Figure 6b shows, the depth of cure of the experimental pit and fissure sealants significantly decreased as the hydrated calcium silicate filler content increased from 5.99 mm to 3.07 mm. This might have been due to the differences in fillers’ refractive indices, especially unsilanized hCS filler decreased the light transmittance of the pit and fissure sealant. According to a previous study, silane treated fillers enhanced light transmission, while uncoated fillers decreased light transmission because of gap formation at the resin matrix-filler interface during light polymerization [44,45]. Per ISO 6874, the depth of cure of pit and fissure sealant material should be greater than 1.5 mm. In the present study, all of the experimental groups met the international standard requirements.

Mechanical and chemical properties of resin composites are related mainly to the water sorption and solubility [46]. In addition, water sorption and solubility are influenced by several factors such as resin matrix or filler components, and proportion of filler in resin composites [47]. Thus, in the present study was performed the water sorption and solubility test to confirm the effect of the calcium silicate material on resin matrix. The release of Ca^2+^ and OH^-^ ions in the oral environment depends on the exposed hydrated calcium silicate surface due to its bioactivity [29]. The experimental groups showed high water sorption and solubility as an increasing amount of hydrated calcium silicate powder was added because resin composites can allow ion release reaction of hCS filler to occur. The acid neutralization and calcium ion release of the experimental materials depended on the water uptake of the hydrated calcium silicate contents through the resin matrix from the aqueous environment. On the other hand, the pit and fissure sealant without hCS filler showed the lowest water sorption and solubility values, due to stable silanized dental glass filler in aqueous environment, which does not cause disintegration compared with the hCS 50.0. It is noteworthy that the hCS 12.5 demonstrated acid neutralizing capability, such as hCS 25.0, hCS 37.5, and hCS 50.0, even though there was no significant difference in water solubility compared to the hCS 0. This can be assumed that the hCS 12.5 has low solubility and feature of continuously preventing dental caries even when the time for exposure in the oral environment is prolonged.

In this study, a novel pit and fissure sealant was developed to inhibit dental caries. This study showed that calcium silicate material has unique and highly desirable acid neutralization properties throughout a prolonged period of time that commercial pit and fissures sealant do not have. However, the resin matrix component used in this study, Bis-GMA, is known as a potentially endocrine interference disruptor and still used as a raw material for dental resin composite materials [48]. Therefore, when developing a resin-based pit and fissure sealant material to be applied to patients based on this study, it is necessary to consider Bis-GMA alternative composition that can be safely used in patients. Another limitation of this study is that we did not assess whether the mechanical properties can be maintained for prolonged period of time similar to those of pit and fissure sealant material currently used in dentistry [7]. Nevertheless, our research indicated that this was the first time a hydrated calcium silicate or non-hydrated calcium silicate filler was shown to rapidly increase the pH of a cariogenic solution. The pit and fissure sealant containing hydrated calcium silicate filler had a high level of initial calcium ion release in a cariogenic environment that otherwise would demineralize the tooth structure [49]. Hence, the pit and fissure sealant containing calcium silicate material can protect tooth structures from demineralization environments. Prior studies reported that hydrated calcium silicate material produced calcium hydroxide, causing the formation of apatite crystallites in biological fluids containing phosphate [50]. In the present study, we could visually observe the appearance of a small amount of deposits on the surface of hCS 50.0 and CS 50.0 specimens, which were taken out after being immersed in distilled water (data not down). Thus, further research is necessary to confirm the clinical efficacy of material for application to tooth structures under simulated microleakage in long-term cariogenic environments. This new hydrated calcium silicate powder has further applications in other dental composites, orthodontic adhesive cement, and adhesives for long-term caries inhibition.

## 5. Conclusions

The present study developed a hydrated calcium silicate dental composite as a pit and fissure sealant with acid-neutralizing and calcium ion releasing properties for caries prevention. The incorporation of hydrated calcium silicate filler in the pit and fissure sealant resulted in a change of the surrounding environment from a cariogenic state into a remineralizing state. In addition, such anti-cariogenic environment around these materials was sustained for a prolonged period of time. Despite deterioration of mechanical and physical properties with addition of hydrated calcium silicate, hCS 12.5 had flexural strength, depth of cure, and water solubility matching the hCS 0 control group. Therefore, newly developed hCS composite is promising for caries-preventing pit and fissure sealants, and its rapid acid-neutralizing and calcium ion release properties may have a wide range of applications in dental materials.

## Figures and Tables

**Figure 1 polymers-12-01200-f001:**
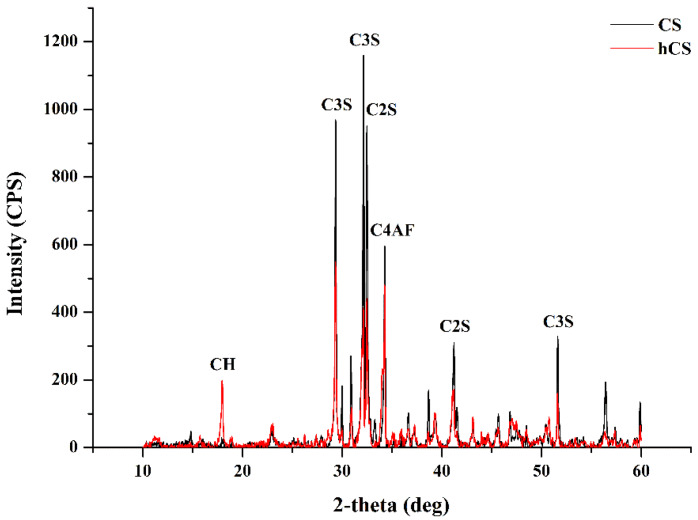
X-ray diffraction patterns of the non-hydrated CS powder (black line) and hydrated CS powder (red line) stored in distilled water. CH, calcium hydroxide; C3S, tricalcium silicate; C2S, bicalcium silicate; C4AF, tetracalcium aluminoferrite.

**Figure 2 polymers-12-01200-f002:**
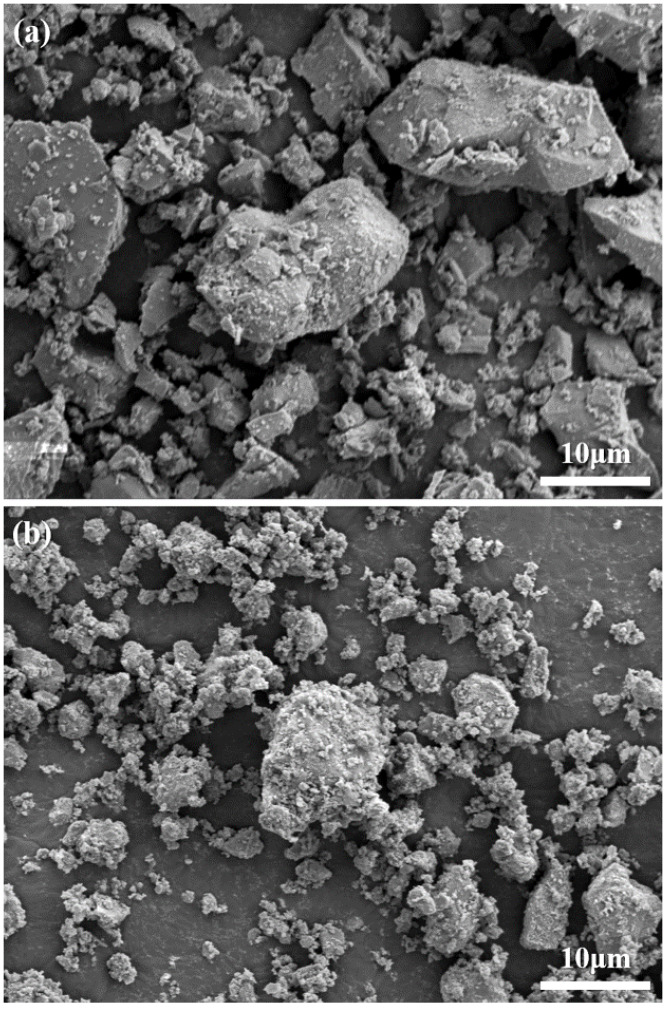
Scanning electron microscopic images of the CS powder before hydration (**a**) and after hydration in distilled water (**b**).

**Figure 3 polymers-12-01200-f003:**
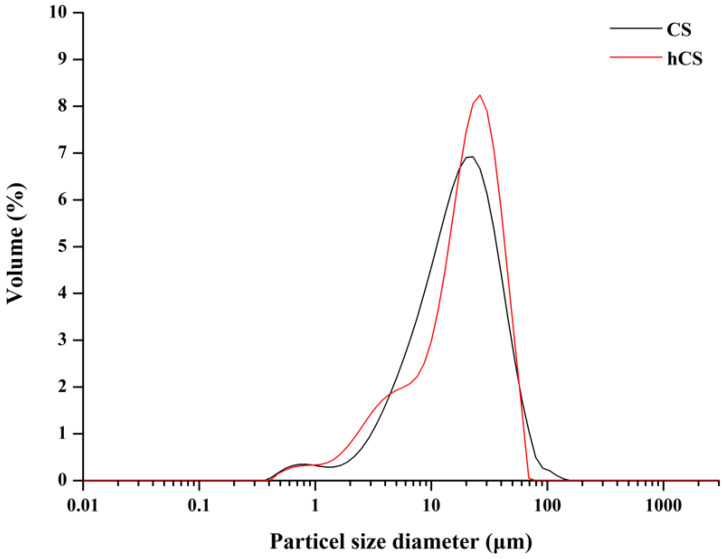
Particle size distribution of the non-hydrated CS powder (black line) and hydrated CS powder (red line) stored in distilled water.

**Figure 4 polymers-12-01200-f004:**
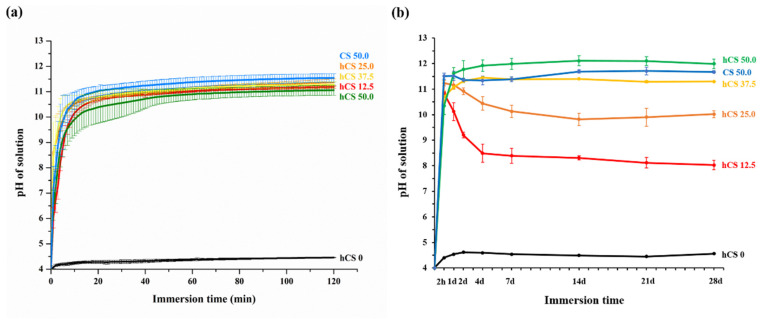
The pH value of the lactic acid solution recorded after the specimens were immersed for 120 min (**a**) and prolonged periods of time from 2 h to 28 d (**b**). Each value represents the average of 5 repeated measurement, and the error bars indicate the standard deviation of the average values (average ± standard deviation; *n* = 5).

**Figure 5 polymers-12-01200-f005:**
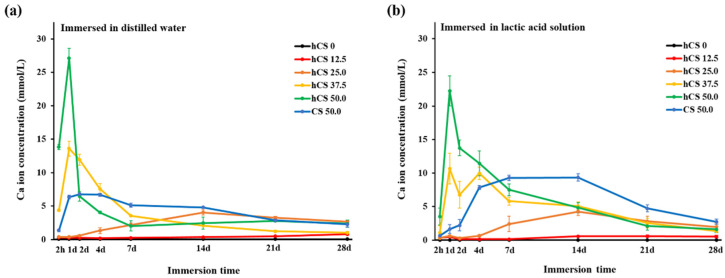
Concentration of Ca ion released into a refreshed solution at each measurement time point from 2 h to 28 d in distilled water (**a**) and lactic acid solution (**b**). Each value represents the average of 5 repeated measurement, and the error bars indicate the standard deviation of the average values (average ± standard deviation; *n* = 5).

**Figure 6 polymers-12-01200-f006:**
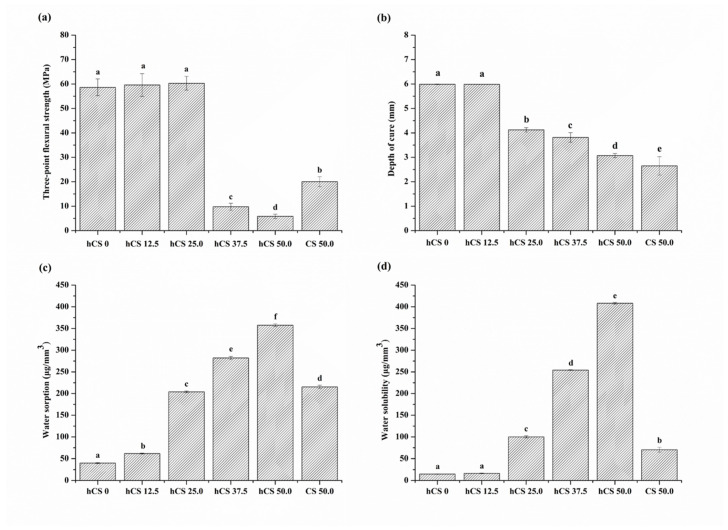
Results of the flexural strength (**a**), depth of cure (**b**), water sorption (**c**), and solubility (**d**) of each group. Each value represents the mean of 6 measurements, and the error bars represent the standard deviation of the mean (average ± standard deviation; *n* = 6). For each of the mechanical and physical properties, the same lowercase letter indicates no differences among the experimental groups (*p* > 0.05).

**Table 1 polymers-12-01200-t001:** Weight ratio of the filler proportions in each group (%).

Group	Resin Matrix	Amount of hCS Filler	Amount of Portland Cement CS Filler	Amount of Silanized Dental Glass Filler
hCS 0	50.0	0.0	0.0	50.0
hCS 12.5	50.0	12.5	0.0	37.5
hCS 25.0	50.0	25.0	0.0	25.0
hCS 37.5	50.0	37.5	0.0	12.5
hCS 50.0	50.0	50.0	0.0	0.0
CS 50.0	50.0	0.0	50.0	0.0

hCS: Hydrated calcium silicate. CS: Portland cement.

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
