# Peer review of "Prevention of Secondary Caries Using Resin-Based Pit and Fissure Sealants Containing Hydrated Calcium Silicate"

_polymers, 2020, doi:10.3390/polym12051200_

Round 1
Reviewer 1 Report
I have some concerns about the present paper.
A Ca-ions biocompatible material can characterize well a suitable route towards improved filler composites. However, I doubt its cements alone can match requirements for applications nowadays, at least for long term occlusions.
I appreciated the discussion on the chemical side of the work, the acid-neutralizing action, the pH compatibility, and calcium ion release. These topics are more than promising and properly presented and discussed in the paper. The concern comes from mechanical properties.
SEM images and Fig.3 in the paper tell us this cement cannot work in composites. In my opinion, it needs for smaller particles (2-10 nm) to fill cavities between large hydroxide (macrofiller) grains and silanized glass particles. Nanoparticles would act as a “liquid” to prevent the occurrence of pits that can be cured hardly by a cement-based composite.
That is, the Vickers figure is minimal for long term applications as compared, for instance, with other fillers, and with natural enamel* notably.
Unfortunately, Bis-GMA is still adopted, although its use should be minimized since it is a resilient endocrine interferent disruptor. Bis-GMA may cause interferences in organogenesis resulting in many pathologies in particular in childhood.
In conclusion, I would suggest the publication of the present work once the Authors could emphasize, beyond the scientific relevance of the cement-based resin, the limits in applications. A comparison with industrial composites is necessary, in my opinion, unrestricted to the internal standards implemented by the Authors. These discussions will make a reader aware of the complexity required by real applications to patients.
*See for instance in table 4 of the chapter “Innovative Fillers For Dental Resin” of the book “Comparative Effectiveness Research (CER): New Methods, Challenges and Health Implications” 2016.
Author Response
|
Comment #1 |
SEM images and Fig.3 in the paper tell us this cement cannot work in composites. In my opinion, it needs for smaller particles (2-10 nm) to fill cavities between large hydroxide (macrofiller) grains and silanized glass particles. Nanoparticles would act as a “liquid” to prevent the occurrence of pits that can be cured hardly by a cement-based composite. *See for instance in table 4 of the chapter “Innovative Fillers For Dental Resin” of the book “Comparative Effectiveness Research (CER): New Methods, Challenges and Health Implications” 2016. |
|
Response |
Thank you very much for your helpful comments. I fully agree with your opinion that it needs smaller particles to fill cavities between large hCS grains and silanized glass particles for improving mechanical properties. Thus, taking your suggestions and the above comments into consideration, the “Discussion” is modified to include the details and references about other studies (Lines 364-367 and 372-376). Thank you once again for your input. |
|
Comment #2 |
Unfortunately, Bis-GMA is still adopted, although its use should be minimized since it is a resilient endocrine interferent disruptor. Bis-GMA may cause interferences in organogenesis resulting in many pathologies in particular in childhood. |
|
Response |
Thank you once again for your valuable comments about Bis-GMA, is known as a potentially endocrine interference disruptor and still used as a raw material for dental resin composite materials. I agree with your comments and “Discussion” is modified to include the details and references (Lines 409-416). Thank you once again for your input. |
|
Comment #3 |
In conclusion, I would suggest the publication of the present work once the Authors could emphasize, beyond the scientific relevance of the cement-based resin, the limits in applications. A comparison with industrial composites is necessary, in my opinion, unrestricted to the internal standards implemented by the Authors. These discussions will make a reader aware of the complexity required by real applications to patients. |
|
Response |
Thank you very much for your helpful comments. According to your advice, the limitations of this study was added and improved the contents with scientific literature in “Discussion” (Lines 372-376 and 409-416). |
Reviewer 2 Report
Thanks for the article, and I have the following comments:
1) The size of fillers are different. (and hCS and commercial filler should also have different strength, right?) So, increasing the amount of hCS would reduce the flexural strength. Please discuss this point. (but, what's the significance of flexural strength in pit and fissure sealant?)
2) Since the fillers have different refractive indices, so the depth of cure would be affected. You can also add this point in discussion.
3) immersion in distilled water - why the pH is 6.5?
4) Is there any disintegration of your materials after prolonged water or acid storage? As you see the solubility increase with of the hCS filler content.
5) Fig 5 - Ca release - is it a cumulative Ca release? Or, is it a quenched data at certain time point? If it is a cumulative Ca, then that means the material will ad/absorb Ca after certain time. How can you explain this phenomena in discussion, together with the pH change? Please add in discussion.
6) Did you observe any crystals on the experimental sealants with hCS or CS after immersion in water?
Author Response
|
Comment #1
|
The size of fillers are different. (and hCS and commercial filler should also have different strength, right?) So, increasing the amount of hCS would reduce the flexural strength. Please discuss this point. (but, what's the significance of flexural strength in pit and fissure sealant?) |
|
Response |
Thank you very much for your helpful comments. In this study, there was a significant difference in flexural strength between hCS 50.0 and hCS 0. Although, as your comment, the size difference between hCS and commercial filler cannot be ruled out, according to previous papers, the silane treatment is an important factor causing the difference in resin composite strength. Therefore, we have included some of your comments in “Discussion” of our manuscript (Lines 364-367). Thank you once again for your input. |
|
Comment #2 |
Since the fillers have different refractive indices, so the depth of cure would be affected. You can also add this point in discussion. |
|
Response |
Thank you very much for your helpful comments. According to your suggestion, we’ve added the statement for the refractive indices in the “Discussion” to clarify the result for depth of cure (Lines 384, 387). |
|
Comment #3 |
Immersion in distilled water - why the pH is 6.5? |
|
Response |
Distilled water used in this study was commercially available distilled water product (JW pharmaceutical, Seoul, Korea) purchased from a pharmaceutical company in Korea, and it was recorded as pH 6.5 at room temperature. Moreover, one study showed that the natural pH of the distilled water is about 6.5 [1]. Therefore, we thought that the use of pure distilled water was appropriate to measure the exact amount of calcium ions released from the specimens, and thus “Materials and Methods” of our study was designed and carried out (Lines 144-145). [1] Tripathy, S.S.; Bersillon, J.L.; Gopal, K. Removal of fluoride from drinking water by adsorption onto alum-impregnated activated alumina. Separation and purification technology. 2006, 50, 310-317. |
|
Comment #4 |
Is there any disintegration of your materials after prolonged water or acid storage? As you see the solubility increase with of the hCS filler content. |
|
Response |
Thank you for providing these insights. Water sorption and solubility are related mainly to the mechanical and chemical properties of the resin composites in an aqueous environment. Moreover, there are several factors that influence water sorption and solubility values, such as resin matrix or filler components, and proportion of fillers in resin composites. The absorbed water in the pit and fissure sealant can allow ion release reaction of hCS filler to occur. However, the pit and fissure sealant without hCS filler showed the lowest water sorption and solubility values, most likely due to the presence of stable silanized dental glass filler, which does not cause disintegration of materials. It is noteworthy that the hCS 12.5 demonstrated acid neutralizing capability, such as the experimental groups with a relatively high hCS ratio, even though there was no significant difference in water solubility compared to the hCS 0. This can be assumed that the hCS 12.5 has low solubility and feature of continuously preventing dental caries even when the time for exposure in the oral environment is prolonged. Therefore, we have included some of your comments in “Discussion” of our manuscript (Lines 390-406). Thank you once again for your input. |
|
Comment #5 |
Fig 5 - Ca release - is it a cumulative Ca release? Or, is it a quenched data at certain time point? If it is a cumulative Ca, then that means the material will ad/absorb Ca after certain time. How can you explain this phenomena in discussion, together with the pH change? Please add in discussion. |
|
Response |
Thank you very much for your helpful question. Figure 5 is not a cumulative Ca release result, but the graphs showing only the quenched concentration of Ca ion released into a refreshed solution at each measurement time point. Also the legend of “Figure 5” is modified to include the details about your valuable question (Lines 258). Thank you once again for your input. |
|
Comment #6 |
Did you observe any crystals on the experimental sealants with hCS or CS after immersion in water? |
|
Response |
Thank you for your valuable question. In the present study, we could visually observe the appearance of a small amount of deposits on the surface of hCS 50.0 and CS 50.0 specimens, which were taken out after being immersed in distilled water. Although the analysis data for the findings were not added to our manuscript, this will be revealed in the further research. Also the “Discussion” is modified to include the details about your valuable question (Lines 423-425). Thank you once again for your input. |
Round 2
Reviewer 1 Report
I acknowledge that Authors have fulfilled completely my concerns, thus I can suggest the publication of the present paper.
A final very minor correction is required on ref 41 since the editor of the book in the reference is Chiappelli F., instead of Francesco C. and the chapter involved in the discussions is the second one.